# The Role of Negative Perfectionism and the Relationship between Critical Thinking and the Halo Effect: Insights from Corporate Managers in Human Resources

**DOI:** 10.3390/bs13070533

**Published:** 2023-06-26

**Authors:** Jiayi Lv, Zhaoyang Sun, Hao Li, Yubo Hou

**Affiliations:** 1School of Psychological and Cognitive Sciences, Peking University, Beijing 100871, China; jiayilv@pku.edu.cn (J.L.); zhaoysun@stu.pku.edu.cn (Z.S.); 2Beijing Key Laboratory of Behavior and Mental Health, Peking University, Beijing 100871, China; 3Plateau Brain Science Research Center, Tibet University, Lhasa 850000, China

**Keywords:** halo effect, critical thinking, negative perfectionism, human resources, compensation

## Abstract

This research aims to explore the relationship between critical thinking and the halo effect among managers working in the Human Resources (HR) departments of corporations. By utilizing a sample of over 301 corporate HR managers as participants, this study provides valuable insight into the dynamics between critical thinking, the halo effect, and the mediating role of negative perfectionism. The findings of this study suggest a significant negative relationship between critical thinking and the halo effect, as well as a significant positive relationship between negative perfectionism and the halo effect. Notably, negative perfectionism acts as a mediator between critical thinking and the halo effect. Our research also reveals that compensation level moderates this relationship, with lower-income HR managers exhibiting a stronger association between negative perfectionism and the halo effect compared to higher-income HR managers. These findings significantly contribute to our understanding of the interplay between critical thinking and the halo effect among HR managers in corporate settings. Identifying negative perfectionism as a mediating factor clarifies the underlying mechanisms between critical thinking and the halo effect, while the moderating effect of compensation level highlights the importance of considering contextual factors. The practical implications of this research include the significance of promoting critical thinking skills among HR managers to mitigate the halo effect in job recruitment and performance evaluation. Additionally, organizations should prioritize fairness and consistency in compensation levels to minimize the influence of negative perfectionism and its impact on the halo effect.

## 1. Introduction

During the process of recruiting suitable job candidates for a corporation, HR managers play a pivotal role. However, several factors influence HR managers when making these crucial hiring decisions. One such factor that significantly impacts HR managers during recruitment is the halo effect. HR managers are responsible for selecting talents, and their effectiveness in this role relies on their ability to identify strengths in individuals and prevent biases influenced by the halo effect. The halo effect, as a cognitive bias, strongly influences HR managers’ talent selection processes [1,2,3]. Within corporate recruitment activities, interviews remain the preferred method for personnel selection among HR managers [4]. Factors such as candidates’ level of education, the prestige of the institution(s) they attended, appearance, eloquence, and versatility are often seen as indicators of ability and competency [5,6]. Careful recruitment is essential to mitigate adverse workplace conditions and reduce employee turnover [7,8]. The halo effect leads HR managers to over-extrapolate and make inaccurate overall assessments based on specific areas where candidates excel. Therefore, it is crucial to address the influence of the halo effect among HR managers during recruitment. How can we mitigate the influence of the halo effect among HR managers to improve their recruitment choices? While previous studies have explored various factors, the examination of thinking styles in relation to enhancing the recruitment process for HR managers is relatively scarce. This research aims to fill the gap by identifying critical thinking as the key to mitigating the halo effect. By promoting critical thinking, HR managers can engage in more rational and unbiased decision-making during recruitment. This study not only complements the existing literature but also provides practical interventions for HR managers to improve their thinking styles and make better choices when selecting the most suitable job candidates for the corporation they work for.

## 2. Literature Review and Hypotheses Development

### 2.1. Halo Effect

The halo effect is a phenomenon in which individuals tend to form an overall impression of others based on positive or negative partial impressions. This bias was discovered by Thorndike in 1920 [9,10], and the study conducted by Dion et al. [11] is widely recognized as a classic contribution to this field. Their research revealed that people have a tendency to overestimate the characteristics and abilities of individuals who they perceive to be more attractive in terms of physical appearance. The halo effect manifests itself in various domains, including in education and judicial practices, wherein biases may influence jury judgments based on the attractiveness of criminal suspects [12] or teachers may form an opinion on a student’s intelligence based on their appearance [13,14]. The emergence of the halo effect is linked to the cognitive mechanism of humans, as impressions of others are formed through a combination of automatic and constructive processes. These processes connect partial characteristics to create a generalized and holistic perception [15,16]. However, the halo effect can be reduced through deliberate and systematic cognitive processing. When individuals perceive things quickly, randomly, and spontaneously, the halo effect can occur. Conversely, when individuals engage in thoughtful and thorough thinking and judgement, the halo effect diminishes [17,18].

### 2.2. Critical Thinking and the Halo Effect

Critical Thinking is a form of reflective thinking that plays a vital role in decision-making processes [19]. Its origin can be traced back to Socrates in ancient Greece, and later, Dewey referred to it as “reflective thinking.” Reflective thinking involves actively and carefully examining one’s beliefs and assumptions, gaining insight into the reasons supporting them and the conclusions they lead to. In the field of psychology, early scholars viewed critical thinking as a cognitive skill that involved information extraction, reasoning, and deduction [20]. Over time, researchers have expanded the concept to include belief, thinking mode, and mental tendencies [16,21]. Recent research has focused on measuring critical thinking. For example, Halpern [22] measured critical thinking based on the ability to analyze, integrate, and evaluate information, as well as the tendency to use these abilities. Byrnes and Dunbar [23] proposed a three-component model of critical thinking, suggesting that critical thinking could be measured from three aspects: critical analytic ability, which involves systematically collecting and analyzing the information related to the problem and evaluating its authenticity; an open-minded view, which reflects an attitude of openness and a willingness to collect and analyze information without limiting oneself to one’s original position; and an effortful view, which entails a desire to do the best one can.

Previous studies have shown that individuals who possess strong critical thinking skills prioritize evidence when making decisions. Such individuals recognize their cognitive limitations and avoid basing decisions solely on personal preferences in the absence of sufficient evidence [24]. Furthermore, they demonstrate respect for diverse perspectives and ideas and strive to learn from them [25]. However, individuals who lack critical thinking skills tend to make decisions based on subjective judgment, disregard the opinions of others, and focus solely on evidence that supports their own views [26]. These individuals also tend to draw one-sided conclusions, lack empathy, and fail to consider alternative perspectives [27], leading to the halo effect. Facione’s self-regulation theory highlights the important role of critical thinking in correcting misconceptions and quickly identifying underlying issues to reduce decision-making errors [23,28]. By avoiding cognitive simplifications, critical thinkers can overcome the halo effect. Therefore, based on these insights, we propose Hypothesis 1: Critical thinking is negatively correlated with the halo effect, suggesting that HR managers with higher levels of critical thinking are less likely to exhibit the halo effect.

### 2.3. The Mediating Role of Negative Perfectionism

Perfectionism is an irrational belief that reflects individual’s cognitive tendency to interpret events and evaluate themselves and others [29]. People with this belief strive for omnipotence and deny their limitations and imperfections [30,31]. For HR managers, perfectionism can have both positive and negative effects [32,33]. Perfectionists usually demonstrate high personal standards, a strong work ethic and a strong sense of morality [17]. However, perfectionists’ high expectations of others often result in harsh evaluations of others’ performance [34]. Moreover, they are prone to forming stereotypes and experiencing the halo effect, as well as negative emotions such as anxiety [35]. To address these contradictions, Slade and Owens [31] proposed a dual process model, which divided perfectionism into positive perfectionism and negative perfectionism. Positive perfectionists uphold high personal standards without excessively worrying about making mistakes. They are often associated with positive psychological variables such as active coping, high self-esteem, achievement, and conscientiousness [36,37]. On the other hand, negative perfectionism, is characterized by an excessive fear of making mistakes, self-doubt, and the belief that personal standards cannot be achieved [38,39]. They tend to make negative judgments when evaluating the ideas of others [40] and pay too much attention to detail. They also tend to filter information and judge others based on specific traits [41]. When HR Managers with negative perfectionism focus on specific traits, they are prone to subjective prejudice and may become caught in a cycle of compulsive thinking and self-argument [34,42].

Critical thinking, as a cognitive skill, enables individuals to analyze the source and the validity of information and set reasonable goals [21,43]. It effectively mitigates the negative effects of perfectionism by helping individuals avoid unrealistically high standards for themselves [31]. When evaluating others, critical thinking can help negative perfectionists adopt a problem-oriented thinking mode [36], thereby reducing cognitive biases caused by negative perfectionism and the halo effect. This leads to the formulation of Hypothesis 2: Negative perfectionism plays a mediation role between critical thinking and the halo effect.

### 2.4. The Moderating Role of Compensation Level

In corporate management, an individual’s compensation serves as both an outcome variable of their performance and an antecedent variable of many job-related psychological behaviors. Compensation level, typically measured by monthly income, is closely related to job performance [44], job satisfaction [45], turnover intention [46,47], and creativity [48] as antecedent variables. Employees’ perception of pay equality and feeling of being underpaid in comparison to others directly impact on their work psychology [22,49]. Moreover, the compensation level of HR managers influences their decision-making, judgment, and behavior during recruitment. According to the resource scarcity theory, a low compensation level often generates a sense of scarcity, leading individuals to focus on maximizing the benefits from their resources [50] and delivering quick results [51,52]. Compared to other departments, HR managers typically have lower compensation levels, which can induce a sense of scarcity and affect their decision-making and judgment [47,53]. Psychologists have found that a lack of money can create a scarcity mindset, diverting limited cognitive resources towards scarce objects and having long-term impacts on cognition and behavior [54]. During recruitment, HR managers are tasked with finding the best candidates [55], which can activate the cognitive model of negative perfectionism [31], depleting cognitive resources and intensifying attentional focus, ultimately resulting in the halo effect. In other words, the low compensation level of HR managers can amplify the impact of negative perfectionism in inducing the halo effect. Therefore, we propose Hypothesis 3: Compensation level moderates the relationship between negative perfectionism and the halo effect. Lower-income HR managers demonstrate a stronger association between negative perfectionism and the halo effect compared to higher-income HR managers.

In summary, this research aims to explore the relationship between critical thinking and the halo effect among HR managers in corporations. Furthermore, it seeks to examine the mediating role of negative perfectionism and the moderating effect of compensation level. The conceptual model is shown in Figure 1.

## 3. Research Method

### 3.1. Participants

Using a simple random sampling method, we collected questionnaires from 350 corporate HR managers from Beijing, Shanghai, Shenzhen, and other locations. After removing participants who had not completed the questionnaire and failed attention checks, 301 valid participants were obtained, resulting in an effective recovery rate of 86.0%. Among them, 98 were male and 203 were female. The average age was 34.77 ± 7.08 years, and the average years of employment were 5.43 ± 5.01 years.

### 3.2. Research Procedure

We developed the online questionnaire using the Questionnaire Star network. The online questionnaire was distributed and collected by the HR departments of the corporations involved in the survey. The data collection process took place over three sessions, with a one-week interval between each session. The participants filled in the questionnaire anonymously. The demographic variables and critical thinking questionnaire were filled out first, followed by the perfectionism questionnaire, and finally, the halo effect questionnaire. After completing the third session, participants received a participation fee of CNY 30.

### 3.3. Research Measures

Background Information. This research collected data on age, gender, level of education, years of employment, and compensation level. The measurement of compensation level takes different forms, and one of the most common forms, monthly income, was used in this study.

Critical Thinking Scale. This research employed a 30-item Chinese Critical Thinking Scale (e.g., “It is interesting to interact with people from different cultures”; see all items in Appendix A) [56], which was developed based on the Facione California Critical Thinking Skills Test (CCTST) and the California Critical Thinking Disposition Inventory (CCTDI) [23]. Each item was rated on a 7-point Likert scale, ranging from 1 (strongly disagree) to 7 (strongly agree). A higher score indicates stronger critical thinking abilities. This scale has been previously employed in studies and has demonstrated favorable consistency [57,58]. The internal consistency of the scale in this study was α = 0.84.

Perfectionism Scale. This research used the Chinese version of the Frost Multidimensional Perfectionism Scale (FMPS; e.g., “If I do not set the highest standards for myself, I am likely to end up a second-rate person”; see all items in Appendix A) [59,60]. This scale consists of 35 items forming six factors: organization, concern over mistakes, personal standards, parental expectations, parental criticism, and doubt about action. Among them, organization indicates positive perfectionism with adaptive significance, while the other factors indicate negative perfectionism without adaptability [55]. Participants rated each item on a 5-point Likert scale, ranging from 1 (disagree) to 5 (agree). A higher score indicates stronger perfectionism. This scale has been utilized in prior studies, exhibiting good consistency [61,62]. In the current sample, the internal consistency was α = 0.84 for positive perfectionism and α = 0.92 for negative perfectionism.

Halo Effect. Using the revised halo effect scale [63], which includes five questions (i.e., “People with a broad face and decent manners are more loyal and reliable”, “People with a good educational background are better learners and perform better at work”, “The outstanding performance of manager’s team shows that the manager is a competent leader with both talent and moral integrity”, “The employees whose job performance is high are also good at interpersonal issues”, and “Neat people are also well-organized”) measuring the degree of the halo effect. Participants rated each item on a 5-point Likert scale, ranging from 1 (disagree) to 5 (agree). A higher score indicates a higher probability of the halo effect occurring during the recruitment process. This scale has been used in previous studies and has shown good consistency [64,65]. The internal consistency of the scale in this study was α = 0.80.

## 4. Research Results

Data were processed using SPSS25.0. To mitigate common method variance, we conducted Harman’s single factor test on our data and employed time-offset multiple data collection. Overall, 11 factors in total were identified to have an eigenvalue greater than 1, explaining 71.02% of the total variance. The top three factors explained 21.81%, 7.29%, and 4.63% of the variance, respectively. None of the factors explained more than 40% of the variance, indicating no significant issues with common method variance. The following analysis aims to explore the relationship between critical thinking and the halo effect among HR managers in corporations.

### 4.1. Correlation Analysis

A simple correlation analysis was first conducted to examine the relationships among the main research variables, and the results are shown in Table 1. Monthly income showed a positive correlation with critical thinking and negative correlations with both positive and negative perfectionism. It also exhibited a negative relationship with the halo effect. Furthermore, critical thinking was positively correlated with positive perfectionism and negatively correlated with negative perfectionism and the halo effect. This suggested that HR managers with higher critical thinking exhibited a lower level of negative perfectionism and a lower level of halo effect, while HR managers with lower critical thinking exhibited a higher level of negative perfectionism and a higher level of halo effect. Additionally, the analysis revealed that positive perfectionism was negatively correlated with the halo effect, and negative perfectionism was positively correlated with the halo effect, which indicated that the higher the negative perfectionism of HR managers, the higher the likelihood of the halo effect in recruitment. These results provided initial evidence for Hypothesis 1 and implied that Hypothesis 2 could hold true. The subsequent regression analysis would further validate Hypothesis 1 and provide evidence for the remaining hypotheses.

### 4.2. Critical Thinking, Negative Perfectionism, and The Halo Effect

We examined the influence of critical thinking on the halo effect and the mediating role of negative perfectionism through a model based on the process provided by Hayes [66]. The results are shown in Table 2. After controlling for age, gender, level of education, years of employment, and monthly income, the overall regression model was found to be significant (*F* (9, 291) =10.040, *R* = 0.486, *R*^2^ = 0.237, *p* < 0.001). Specifically, the total effect of critical thinking on the halo effect was significant (*B* = −0.110, *SE* = 0.024, *t* = −4.95, *p* < 0.001), providing support for Hypothesis 1. Additionally, the negative correlation between critical thinking and negative perfectionism was significant (*B* = −0.512, *SE* = 0.078, *t* = −6.497, *p* < 0.001). The positive correlation between negative perfectionism and the halo effect was significant (*B* = 0.084, *SE* = 0.017, *t* = 4.871, *p* < 0.001).

Using Bootstrap sampling with 5000 iterations, we found that the 95% confidence interval (CI) of the indirect effect did not include zero (*B* = −0.044, *SE* = 0.012, 95%CI [−0.0641, −0.0182]). Moreover, the direct effect of critical thinking on the halo effect was significant (*B* = −0.067, *SE* = 0.025, *t* = −2.694, *p* = 0.007). These findings suggested that negative perfection partially mediated the relationship between critical thinking and the halo effect. Critical thinking migrated the halo effect through reducing negative perfectionism, thus verifying Hypothesis 2.

### 4.3. Moderating Effect of Compensation Level

To examine the hypothesis that compensation level moderates the relationship between negative perfectionism and the halo effect, we conducted additional tests using Model 14 from the Process macro [57], with monthly income as the moderator. After controlling for age, gender, level of education, and years of employment, the overall regression model was found to be significant (*F* (9, 291) = 10.257, *R* = 0.491, *R*^2^ = 0.241, *p* < 0.020). Furthermore, the interaction effect of the monthly income and negative perfectionism significantly predicted the halo effect (*B* = −0.061, *SE* = 0.007, *t* = −2.492, *p* = 0.010), which suggested that the compensation level moderated the relationship between negative perfectionism and the halo effect. Specifically, when the compensation level was lower (less than CNY 8000 per month), negative perfectionism had a significant positive impact on the halo effect (*B* = 0.139, *SE* = 0.022, *t* = 6.231, *p* < 0.001). Conversely, when the compensation level was higher (more than 15,000 RMB per month), negative perfectionism had no significant influence on the halo effect (*B* = 0.064, *SE* = 0.022, *t* = 2.909, *p* = 0.301). These results supported Hypothesis 3, indicating that compensation level moderated the influence of negative perfectionism on the halo effect.

## 5. General Discussion

This research explores the influence of critical thinking on the halo effect among HR managers, with a specific focus on the mediating role of negative perfectionism and the moderating role of compensation level. The findings suggest that critical thinking is negatively correlated with the halo effect. Critical thinkers tend to employ detailed analysis and intricate logic when confronted with problems, making them less susceptible to the biases caused by the halo effect. Mercier and Sperber propose that critical thinkers maintain a constant state of epistemic vigilance [67]. Ennis also listed “seek and offer clear statements of the conclusion or question” as one of the general critical thinkers’ dispositions [68]. Moreover, the research demonstrates a positive correlation between negative perfectionism and the halo effect. Individuals with higher levels of negative perfectionism are more prone to showing the halo effect. Negative perfectionists often have high standards for themselves or others, fearing mistakes and criticism from others. As a result, their cognition and attention become more narrow and focused, leading them to overlook valuable information and making them more vulnerable to the halo effect when evaluating others [69].

Mediation analysis reveals that negative perfectionism mediates the relationship between critical thinking and the halo effect. Specifically, critical thinking is found to reduce negative perfectionism, which consequently diminishes the halo effect. As mentioned earlier, negative perfectionism has been shown to be positively associated with depression and anxiety, while negatively correlated with self-esteem and life satisfaction [70,71]. Neuroscience research provides evidence that negative emotions activate the right brain hemisphere for advanced cognitive tasks [72]. The right hemisphere is responsible for facial perception and emotional function, with facial recognition information originating in the right hemisphere and being transmitted to the left. Although the left hemisphere also plays a role in facial recognition, it is less efficient at retrieving such information [73]. This neural mechanism leads to negative perfectionists relying more on emotional thinking than logical reasoning or rational analysis during recruitment and candidate selection tasks, which is a key factor contributing to the halo effect. Negative perfectionism can lead individuals to form irrational beliefs and emotions [27], which can adversely impact their cognitive abilities and behaviors [74]. Critical thinking can reverse the irrational pattern of negative perfectionism and reduce the incidence of the halo effect.

The present study also found that the compensation level of HR managers plays a moderating role in the pathway from negative perfectionism to the halo effect. Specifically, the effect of negative perfectionism on the halo effect is significant when one’s monthly income is considered to be low (less than CNY 8000 per month), while it has minimal impact among those whose monthly income is considered to be high (more than CNY 15,000 per month). Why does income level serve such a role? Mani et al. [50] offered a potential explanation for the relationship between poverty and cognitive tendencies. They suggested that poverty simplified the thinking process of HR managers, making them more prone to the halo effect. This mediating effect may result from the narrowed focus of attention caused by resource scarcity, such as a lack of money, which can impair cognitive judgment [46]. The resource limitation theory also suggests that human attention is limited, and more cognitive resources are required when managing complex stimuli or tasks [51]. Consequently, individuals with low income tend to concentrate on specific aspects and process superficial information when addressing tasks, which makes them more vulnerable to the halo effect [75].

### 5.1. Theoretical Implications

This research holds theoretical implications by enriching the current knowledge regarding the relationship between critical thinking and the halo effect among HR managers in corporate settings. By emphasizing the negative relationship between critical thinking and the halo effect, our research complements the theoretical framework surrounding these constructs. Moreover, the identification of negative perfectionism as a mediator provides valuable insights into the underlying mechanism involved in this relationship. This research also extends the theory by discovering the moderating effect of compensation levels between critical thinking and the halo effect. This reveals a deeper understanding of the contextual factors that shape HR managers’ decision-making processes. These theoretical implications contribute to a more comprehensive understanding of critical thinking and the halo effect dynamics in organizational contexts.

### 5.2. Practical Implications

The research findings hold significant implications for the selection, training, and development of HR managers, as well as their compensation systems. First, fostering critical thinking skills among HR managers can effectively reduce the halo effect in recruitment and performance evaluation processes. Enhancing their critical thinking abilities equips HR managers to make more informed and impartial decisions [76]. Second, the findings suggest the need for organizations to be aware of the potential impact of compensation levels on HR managers’ judgement and decision-making, ensuring fairness and consistency to minimize the influence of negative perfectionism and its amplifying effect on the halo effect.

### 5.3. Limitations and Future Directions

This research has several limitations that need to be acknowledged. First, the structural model derived from the questionnaire-based survey method may require further validation in future studies, and efforts should be made to improve its ecological validity. Second, the relationship between variables used in this research demands further consideration. For instance, the compensation level examined in the research might be correlated with factors such as age, seniority, and length of employment. It is essential to explore how these relationships influence the findings. Lastly, the model identified in this research may need to be evaluated within a broader context, taking into account not only internal and external factors but also their interactions with one another.

For future research, it is recommended to include a broader range of participants from diverse industries and organizations to enhance the external validity of the findings. Exploring additional variables, such as cognitive biases or decision-making styles, can deepen our understanding of the recruitment process and its intricate relationship with critical thinking. Acting upon these research directions will help to advance of knowledge and further refine practical interventions in improving recruiting processes.

## 6. Conclusions

Based on our research, we have drawn the following conclusions: (1) Critical thinking can effectively reduce the halo effect experienced by HR managers. (2) Negative perfectionism mediates the relationship between critical thinking and the halo effect, explaining the driving mechanism. (3) The association between negative perfectionism and the halo effect is moderated by an individual’s compensation level, specifically observed among those with lower compensation level but not among those with higher compensation levels. These findings significantly contribute to our understanding of the interplay between critical thinking and the halo effect among HR managers in corporate settings. Moreover, this research holds practical implications by demonstrating the significance of promoting critical thinking skills among HR managers and by inspiring organizations to prioritize fairness and consistency in compensation levels to minimize the influence of negative perfectionism and its impact on the halo effect.

## Figures and Tables

**Figure 1 behavsci-13-00533-f001:**
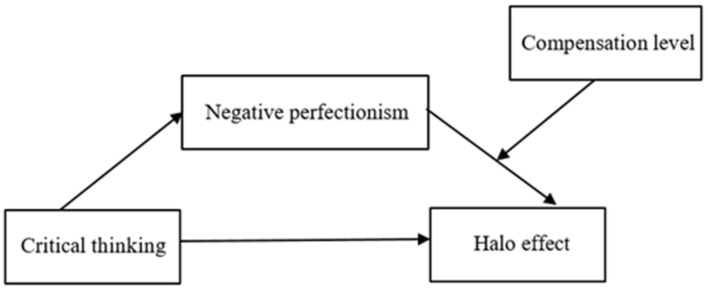
Conceptual model.

**Table 1 behavsci-13-00533-t001:** Variable mean, SD, and correlation matrix (*n* = 301).

Variables	M	SD	1	2	3	4	5
Monthly income	4.91	2.43	0.616 **				
Critical thinking	2.08	0.250	0.291 **	0.321 **			
Positive perfectionism	3.91	0.670	0.058	−0.123 *	0.147 *		
Negative perfectionism	2.92	0.640	−0.178 **	−0.221 **	−0.406 **	0.180 **	
Halo effect	4.54	1.10	0.002	−0.185 **	−0.257 **	0.248 **	0.339 **

Note. * *p* < 0.05, ** *p* < 0.01.

**Table 2 behavsci-13-00533-t002:** The main effect and the mediating effect (*n* = 301).

Variables	*B*	*SE*	*t*	*p*
Control variables				
Gender	−0.763	0.629	−1.212	0.226
Age	0.157	0.055	2.817	0.005
Level of education	−0.545	0.445	−1.226	0.221
Years of employment	0.939	0.078	1.212	0.227
Monthly income	−0.634	0.178	−3.562	<0.001
Father’s level of education	0.274	0.285	0.965	0.335
Mother’s level of education	−0.423	0.307	−1.376	0.169
Positive perfectionism	0.252	0.074	3.382	0.008
The effect of Independent variable on intermediate variable (a)				
Critical thinking -> Negative perfectionism	−0.512	0.078	−6.497	<0.001
Direct effect of intermediate variable on dependent variables (b)				
Negative perfectionism -> Halo effect	0.084	0.017	4.871	<0.001
The Effect of independent variable on dependent variable				
Total effect (c)	−0.110	0.024	−4.953	<0.001
Direct effect (c′)	−0.067	0.025	−2.694	0.007
Indirect effect (ab)	−0.044	95%CI = [−0.0641, −0.0182]		

## Data Availability

The data presented in this study are available upon request from the corresponding author.

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
