# Peer review of "The Role of Negative Perfectionism and the Relationship between Critical Thinking and the Halo Effect: Insights from Corporate Managers in Human Resources"

_behavsci, 2023, doi:10.3390/bs13070533_

Round 1
Reviewer 1 Report
Interesting Topic and accepted with minor changes.
1) Abstract: it is too short extend it and add some implications
2) Introduction: Authors should explain or introduce research, i.e., starting with:
Research Gap
Why this study is important and needful for Management in Practice and Theory
Don't start with Heading like:
1.1. Halo Effect
Its seems like Literature review
Add: Section 2 namely Literature Review
Also add Figure Conceptual Model in LR section
3) Methodology shows only: sampling method indicate compelete research design?
How data collection made?
You collected personaly or via email?
In research measure show all of items/question in each scale
Also explain who previously used such scales?
Results section is ok
Extend conclusion and add paragraph for each section
1) Implications for theory
2) Implications for practice
3) Limitations and future directions
Idea is great just improve your research so readers can get benefits
Check some strandard publications for top impact journals on same topic for improvement
Can be improved
Author Response
Response to Reviewer 1 Comments
Dear reviewer,
We sincerely appreciate your valuable comments and suggestions on our manuscript “The Role of Negative Perfectionism and Compensation between Critical Thinking and the Halo Effect: Insights from Corporate Human Resource Managers” (behavsci- 2450566). Based on your comments, we have made minor revisions to our manuscript as described below.
Point 1: Abstract: it is too short extend it and add some implications
Response 1: Thank you for your comment and suggestion. We have extended the abstract and added the theoretical implications and the practical implications.
Point 2: Introduction: Authors should explain or introduce research. i.e. strating with Research Gap. Why this study is important and needful for Management in Practice and Theory. Don't start with Heading like: 1.1. Halo Effect, its seems like Literature review. Add: Section 2 namily Literature Review. Also add Figure Conceptual Model in LR section.
Response 2: Thank you for your comment and suggestion. We added an introduction section in the manuscript. In the introduction section we introduced the research and noted the research gap in this field and stated the reasons why we think this study is important and needful for management in theory and practice. Moreover, we changed the title of section 2 to “Theoretical background and Hypotheses” and corrected 2.1 to “Halo Effect Literature Review”. Finally, we added figure conceptual model in the literature review section.
Point 3: Methodology shows only: sampling method indicate compelete research design? how data collection made? you collected personaly or via email? in research measure show all of items/question in each scale. also explain who previously used such scales?
Response 3: Thank you for your comment and suggestion. We have supplemented how the questionnaire was made and how the data was collected. Moreover, we have included all the items of each scale in the main manuscript or in the supplementary material and also explain who previously used such scales.
Point 4: extend conclusion and add paragraph for each section. 1. implications for theory. 2. implications for practice. 3. limitations and future directions.
Response 4: Thank you for your comment and suggestion. In the general discussion section, we have further enriched these contents and additionally added paragraphs for each section: implications for theory; implications for practice and the limitations and future research directions. In addition, we also summarize our results, theoretical contributions and practical contributions in the conclusion section.
Point 5: Comments on the Quality of English Language: can be improved
Response 5: Thank you for your comment and suggestion. We have modified the manuscript overall. We corrected the grammar errors, changed inappropriate use of words and clarified consuming statements in this manuscript..
We sincerely appreciate your valuable comments and suggestions. Should our revisions be inadequate or incorrect, we kindly ask for the opportunity to further revise the manuscript please. Your feedback is invaluable, and we are grateful for your assistance and support throughout this process. Thank you once again for your guidance and support.
Reviewer 2 Report
The article addresses an interesting topic and seeks to explore a relationship between critical thinking and the halo effect among corporate Human Resources, using a sample of over 301 corporate HR managers as participants. The conceptual model is presented and justified.
However, I recommend an update of the literature review (it has only one reference from 2022 and another from 2021).
Given that the scales used (Critical Thinking Scale; Perfectionism Scale; Halo effect scale) have been used in previous studies, it would be interesting to update the literature review.
Author Response
Response to Reviewer 2 Comments
Dear editor and reviewer,
We sincerely appreciate your valuable comments and suggestions on our manuscript “The Role of Negative Perfectionism and Compensation between Critical Thinking and the Halo Effect: Insights from Corporate Human Resource Managers” (behavsci- 2450566). Based on your comments, we have made minor revisions to our manuscript as described below.
Point 1: I recommend an update of the literature review (it has only one reference from 2022 and another from 2021).
Response 1: Thank you for your comment and suggestion. We updated the literature review with 9 more references, including two references from 2023, one reference from 2022, three references from 2021, three references from 2020. These paper were added to the literature review and method section.
We sincerely appreciate your valuable comments and suggestions. Should our revisions be inadequate or incorrect, we kindly ask for the opportunity to further revise the manuscript please. Your feedback is invaluable, and we are grateful for your assistance and support throughout this process. Thank you once again for your guidance and support.